# Effects of physical, psychosocial and dual-career loads on injuries and illnesses among elite handball players: protocol of prospective cohort study

Kristina Drole ,[1] Armin Paravlic,[1,2] Kathrin Steffen,[3] Mojca Doupona[1]

¹Institute of Kinesiology, Faculty of Sport, University of Ljubljana, Ljubljana, Slovenia
²Faculty of Sport Studies, Masaryk University, Brno, Czech Republic
³Oslo Sports Trauma Research Center, Norwegian School of Sport Sciences, Oslo, Norway

**Correspondence to**
Kristina Drole;
kristina.drole@fsp.uni-lj.si

## ABSTRACT

**Introduction** Health problems in sport cause a major burden on several pillars: sport clubs, health and insurance system and mostly the individual athlete. There is limited knowledge in supporting dual-career athletes firmed on evidence-based research in injury/illness prevention, load and stress management. The main goal of this research approach is to determine how specific physical, psychosocial and dual-career loads affect the occurrence of injuries and illnesses in elite handball players and how much of a variation in the athlete's load leads to an occurrence of an injury/illness. A secondary aim is to determine the association between objective and subjective measures of stress as well as examine the benefits of certain biomarkers to monitor stress, load and injury/illness occurrence in athletes.

**Methods and analysis** This prospective cohort study, as part of a PhD project, will be carried out on 200 elite handball players of first men's handball league in Slovenia during an entire handball season, lasting from July 2022 to June 2023. Primary outcomes, including health problems, loads and stress, will be assessed weekly on a player level. Other player-related outcomes will include anthropometry, life event survey and blood biomarkers (cortisol, free testosterone and Ig-A), which will be taken three to five times across the observation period according to the players' training cycle.

**Ethics and dissemination** The project was approved by the National Medical Ethics Committee of Slovenia (number: 0120-109/2022/3) and will be conducted in compliance with the most recent version of Helsinki Declaration. The study results will be published as peer-reviewed articles, congress presentations and as a Doctoral thesis. The results will not only be of importance for the medical and sports community for development of new injury prevention and rehabilitation strategies but also for structuring the correct policy recommendations for athletes' general health.

**Trial registration number** NCT0547129.

## STRENGTHS AND LIMITATIONS OF THIS STUDY

⇒ This longitudinal study will investigate the associations between different types of load and injury/illness occurrence in elite handball players.
⇒ The study will implement holistic monitoring of athlete's health with objective and subjective measures.
⇒ The association between objective and subjective measures of stress in athletes will be investigated.
⇒ The study focuses exclusively on elite male handball players, which can be seen both as a strength and also as a limitation when it comes to generalisability.
⇒ The work has direct implications for sports/clinical practice; periodical athlete monitoring could be beneficial in terms of preventing health issues, lowering health system costs, supporting dual career of athletes, structuring sound policies and enhancing both sports and academic performance.

## INTRODUCTION

Regular moderate exercise is known to have positive health benefits[1]; from preventing a number of non-communicable diseases to improving cardiorespiratory fitness and consequently decreasing cardiovascular morbidity and mortality.[1–3] Top-level sports, on the other hand, are associated with an increased risk of illness and injury.[4 5] There has been a number of studies[6–12] showing a high prevalence of health issues that include injuries and illnesses among elite athletes, ranging from 20% in athletes of all ages[4] to 45% in youth athletes.[5] The injuries and illnesses are not only limiting athletes for the time being[13] but also represent a major burden for sport clubs, health and insurance system and can as well be the reason for sports' career termination.[14–18]

### Athlete's stressors

Sports and non-sports loads pose stress on athletes, leading their physical and psychological well-being from homeostasis to acute fatigue, over-reaching, overtraining, subclinical immune changes, clinical symptoms, illness/injury and possibly death.[19] Research points out that inadequate load management is an important independent risk factor for the development of acute illness and overtraining syndrome.[9 19 20] However, it is not

**Table 1** Questionnaires and tests

| Test/questionnaire | Outcome | Purpose |
|---|---|---|
| SDSCQ | Age, sex, level of education, sport, years of training, player position, previous injuries | General information about the athlete |
| Anthropometry | Body height (cm), body weight (kg) | Measure of basic anthropometric characteristics of an athlete |
| LESCA | Major life events score | Measure of life stress |
| OSTRC | Weekly health problems (acute and overuse injuries, illnesses), measured as prevalence and incidence and severity as OSTRC-severity score and time-loss days | Identifying health problems |
| Weekly load and stress | | Monitoring of athlete's load |
| | Training load in hours | |
| | Competition load in minutes | |
| | Academic load in hours | |
| | Work load in hours | |
| | Average daily sleeping hours | |
| | Perceived stress on scale 1–10 | |
| Blood biomarkers | | Monitoring of stress/insufficient recovery and/or health problem risk |
| Cortisol | Cortisol values in nmol/L | |
| Free testosterone | Free testosterone values in pmol/L | |
| Ig-A | Ig-A values in g/L | |
| | Free testosterone: cortisol ratio | |

LESCA, Life Events Survey for Collegiate Athletes; OSTRC, Oslo Sports Trauma Research Center Questionnaire on Health Problems; SDSCQ, Sociodemographic and Sports-Career Related Questionnaire.

yet possible to quantify how much an increase in training load leads to an increased risk for a given illness[19] or injury.[21] Furthermore, the literature states that there is a risk of illness when an athlete is exposed to other lifestyle stressors that affect the immune system, including sleep deprivation or severe psychosocial stress,[22] such as major life events.[23] A special group of athletes, that might be at higher risk for health problems, are dual-career (DC) athletes. They are posed to the challenge of combining their education/work with elite sport, suggesting their overall weekly load is higher compared to non-athlete students and athletes who are not in the education process. It was found that the highest levels of stress and illness incidence in student-athletes arise during the academically stressful periods.[24 25] Thus, the relationship between stress and physical health is especially relevant for DC athletes, who often face multiple physical and psychosocial stressors (such as transitioning between schools/campuses and home life, frequent travel, academic load and sports load).[26–28]

### Health monitoring and load management

One potential strategy for injury and illness prevention in elite sports is monitoring of external and internal loads. Monitoring an athlete's workload has a number of potential benefits, including interpreting performance changes, increasing understanding of exercise responses, detecting fatigue and monitoring recovery needs,

optimisation of the training programme and competition calendar. It also serves for providing adequate workload levels to reduce the risk of injury and illness[29 30] and guide rehabilitation programmes. External and internal loads can be assessed with subjective and objective measures. External load measurement typically involves quantifying an athlete's training or competition loads by reporting training hours, running length, amount of weight lifted, number of games played and other external factors such as life events, everyday problems and/or travel. On the other hand, internal load is measured by assessing the internal biological, physiological and psychological response to an external load[31] such as heart rate or ratings of perceived exertion. A recent systematic review of internal stress monitoring has identified subjective reporting to be more sensitive and consistent than objective measurements in determining acute and chronic changes in an athlete's well-being.[32]

### Biomarkers

Biomarker discovery and validation is a critical aim of the medical and scientific community.[33] Biomarkers can be used as an objective assessment tool for health, performance, recovery and talent identification in elite sport. Among the most critical hormones for athletes seem to be cortisol, testosterone, dehydroepiandrosterone and immunoglobulin-A (Ig-A). However, testosterone to cortisol (T:C) ratio is considered to be more sensitive

to training stress than either measure alone,[33] as several studies report poor performance outcomes and suboptimal training adaptations in athletes with a low T:C ratio.[34 35] However, to the best of the authors' knowledge, no previous work has examined the use of biomarkers for injury prevention, recovery after an injury or identification of individuals at increased risk.

Given all of the above, there is a need for better health surveillance, load monitoring and support services with an emphasis on injury/illness prevention and rehabilitation.[36] As DC athletes have a wider variety of loads that pose greater psychophysiological strain to them, they could also be at an increased risk for injury/illness. Therefore, more comprehensive monitoring of potential risk factors in this specific population is warranted. It is not yet known, but important to determine how much of overall load can an athlete withstand before injury/illness occurs. Thus, the primary aim of this study is to determine how specific physical, psychosocial and DC loads affect the occurrence of injuries and illnesses in elite handball players and how much of a variation in the athlete's load leads to an occurrence of an injury/illness. A secondary aim is to determine the association between objective and subjective measures of stress as well as examine the benefits of certain biomarkers to monitor stress, load and injury/illness occurrence in athletes.

## METHODS AND ANALYSIS
### Study design
The study is designed as a prospective cohort study. All measurements will be conducted over a single handball season, lasting from July 2022 to June 2023 (a total of 46 weeks), with weekly reporting (table 1) and three to five biomarker sampling timepoints (figure 1). We used the Standard Protocol Items: Recommendations for Interventional Trials (SPIRIT) checklist when writing our report.[37]

### Participants and sample size
A power analysis was a priory determined using the G*Power software.[38] Based on previous studies with similar design, we expected to find nearly two times as high odds of injury occurrence in athletes during a period of high academic stress compared with low academic stress (OR=1.78; effect size (ES)=0.14)[25] with power of 0.90 and α=0.05, two-tailed, which calculated an initial sample size of 134 participants. As in sports practice, the athlete drop-out is to be expected (due to travelling, international transfer to another club/state, etc); we adjusted the originally calculated sample size by following formula: N1=n/(1−d),[39] where N1 is adjusted sample size, n is the sample size required as per the proposed formula (N=134) and d is the drop-out rate (d=0.10). This resulted in total sample size of 149 athletes. Inclusion criteria will be as follows: DC and non-DC registered male handball players who are competing in the first Slovenian handball league and are above 18 years of age.

### Assessment and outcomes
#### Oslo Sports Trauma Research Centre Questionnaire on Health Problems
The Oslo Sports Trauma Research Centre Questionnaire on Health Problems (OSTRC) is a tool to capture and monitor health problems longitudinally in athletes. It consists of 18 questions, which are answered for the past 7 days. The first four questions are to determine the severity of the health problem, while the rest are to further classify the health problem, such as type and anatomical location of the problem.[40] As per previously established protocol, the athletes in our study will self-report their health status weekly with the help of a physiotherapist,

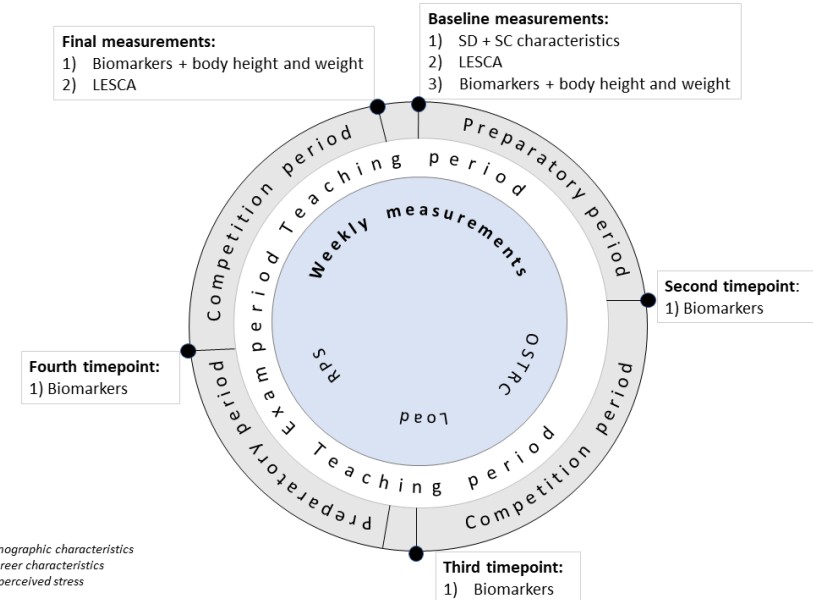

**Figure 1** Assessment plan. LESCA, Life Events Survey for Collegiate Athletes.

while using the OSTRC questionnaire.[40] The athletes with reported health problem will be contacted by a medical professional to clinically verify the reported health case.

## Life Events Survey for Collegiate Athletes Questionnaire

At the baseline and final measurements, the athletes will be asked to complete Life Events Survey for Collegiate Athletes (LESCA)[41] in order to get more contextualised view of the life events that were happening in the past 12 months. LESCA is a 69-item questionnaire where athletes rate how each life event affected their lives in the past 12 months on a −4 to 4-point scale (−4, extremely negative to +4, extremely positive). The events include personal, academic-related and sports-related events. From summing the rated life events, two life stress scores, negative (NEG) and positive (POS) are obtained, while a total life stress score can as well be obtained by summing the NEG and POS scores together.[41]

## Weekly load and stress questionnaire

The teams will weekly report training and competition load for each athlete, while academic (educational activities, including lectures, exams and studying) and work load (a part-time or full-time job besides playing sports) for the past 7 days will be reported by the athletes themselves. Competition load will be reported in minutes for each game played in the past week, while training, academic and workload will be reported in hours. The athletes will additionally report average hours slept per day over the past week and rate their perceived overall stress of that week on the scale 1–10.

## Blood biomarkers

There will be three to five measurement timepoints for biomarker sampling, according to the training (preparatory period, competitive period, off-season period) cycle. The athletes will be tested under similar conditions every time, after a rest day (refrainment from intensive training the day before blood collection), between 7:00 and 10:00 in the morning.

## Statistical analysis

Statistical analysis will be performed using standard statistical software (SPSS V.27.0, IBM, Armonk, New York). Descriptive statistics will be used for all variables. Normality of data distribution will be assessed using the Shapiro-Wilk test. In case of non-normally distributed data, non-parametric statistics will be used, otherwise, the statistics for normally distributed data will be used. The data will be analysed using a logistic regression model to properly account for the dependency induced by the repeated observations over time within each subject. Main effect of all dependent variables on injury/illness occurrence will be studied using repeated measures analysis of variance (ANOVA) with time (preparation period vs competition period vs off-season period) as within factor, while group (injured/ill vs healthy) as between factor. Where significant effects will be found for group, time effects or two-way interactions, pairwise comparisons

will be used to address the significant differences for each variable independently. For comparison of the groups, we will use independent t-test. The magnitude of effect for all dependent variables will be reported as partial eta squared ($\eta^2$). In addition, *Cohens' d* (ES) will be used to assess magnitude of difference between groups. Furthermore, regression analysis will be used to assess predictive power of individual parameters to identify injury/illness occurrence. For the correlation of subjective measures of stress with the objective measures, bivariate correlation will be used. Additionally, Bland-Altmann analysis will be used (low/high stress normative values) as previously recommended.[42] Furthermore, receiver operating characteristic (ROC) curve analysis will be used for defining the cut-off values. Hypotheses will be accepted or rejected at $p \leq 0.05$.

## Patient and public involvement

Patients and/or the public were not involved in the design and/or conduct, and/or reporting of this study. The project results will be presented to the participants in collaboration with the team staff (strength and conditioning coaches, physiotherapists, kinesiologists, etc). Furthermore, the participants of the proposed study will be invited to share their experiences of their involvement in our project with elite athletes playing other sports in the Republic of Slovenia in order to continue to implement 'good practice' with future projects.

## Ethics and dissemination

The project was approved by the National Medical Ethics Committee of Slovenia (number: 0120-109/2022/3) and registered on Clinical Trials.gov (registration number: NCT05471297). The study will be conducted in compliance with the most recent version of Helsinki Declaration of 1975 and the American College of Sports Medicine Policy Statement Regarding the Use of Human Subjects and Informed Consent. Any changes made to the study protocol will be submitted to the National Medical Ethics Committee for consideration and approval. Participants will be introduced to the aims and objectives of the study, noticed that participation in the study is voluntary and asked to sign an informed consent, collected by KD. Consent document also states that the participant is free to withdraw from the study without impact on their future status in the team and without having to provide the reason for their withdrawal. Data on each participant will be anonymised using only ID numbers. The study will be coordinated by KD, a PhD student conducting her PhD thesis on this topic. As such, the study results will be published as peer-reviewed articles, conference presentations and, finally, as a Doctoral thesis.

**Acknowledgements** The authors wish to thank University Medical Centre Ljubljana, Clinical Institute of Clinical Chemistry and Biochemistry for supporting the upcoming blood biomarker analysis.

**Contributors** KD contributed to the idea, conceptualisation and drafted the manuscript; KD and AP contributed to design and planned analysis of the study.

AP, KS and MD critically revised the manuscript. KD, AP, KS and MD gave their final approval for submission.

**Funding** The work is supported by a research fellowship grant (number 393/2020) received by KD from the Public Research Agency of the Republic of Slovenia (ARRS). The research took place within the Kinesiology of Monostructural, Polystructural and Conventional Sports research program, code: P5-0147, financed by the Public Research Agency of the Republic of Slovenia. The funding agency has no impact on data collection, analysis or interpretation of the study results.

**Competing interests** None declared.

**Patient and public involvement** Patients and/or the public were not involved in the design, or conduct, or reporting, or dissemination plans of this research.

**Patient consent for publication** Not applicable.

**Provenance and peer review** Not commissioned; externally peer reviewed.

**ORCID iD**
Kristina Drole http://orcid.org/0000-0002-8403-3154

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
