## [Reviewer comments · BMJ Open]

ARTICLE DETAILS

TITLE (PROVISIONAL)	Effects of physical, psychosocial and dual-career loads on injuries and illnesses among elite handball players: protocol of prospective cohort study
AUTHORS	Drole, Kristina; Paravlic, Armin; Steffen, Kathrin; Doupona, Mojca

VERSION 1 – REVIEW

REVIEWER	Gregov, Cvita University of Zagreb
REVIEW RETURNED	02-Nov-2022

GENERAL COMMENTS	The protocol of the study is very interesting and hopefully useful in determining the influence of psychosocial and physical loads on injuries and illnesses in athletes' dual career. I congratulate the authors and have no further comments on the manuscript's content.
--

REVIEWER	Bjørndal, Christian
REVIEW RETURNED	22-Nov-2022

GENERAL COMMENTS	Thank you for the opportunity to review your study protocol. I believe your study show potential to become a significant contribution to the research
---

REVIEWER	Rogers , Bruce University of Central Florida, Internal Medicine
REVIEW RETURNED	13-Dec-2022

GENERAL COMMENTS	Intro Line 78, remove “although”, otherwise it does not make sense. Line 116 – rate to ratings Stats Make sure data is normally distributed to qualify for those selected tests. If not normal an entirely different set of comparisons are needed. With Bland Altman analysis, the differences need to be normally distributed. You should also check for proportional bias using regression-based calculation of mean differences and limits of agreement (Ludbrock, Clin. Exp. Pharmacol. Physiol. 2010, 37, 143–149). The issue of “free testosterone” Unless you are measuring free testosterone by equilibrium dialysis (doubtful) I would recommend you measure total T and SHBG per most guidelines (https://onlinelibrary.wiley.com/doi/10.1111/cen.13888 and
---

	https://www.nature.com/articles/s41443-019-0144-9). You will still be able to derive a “calculated” FT (http://www.issam.ch/freetesto.htm). Additional problems with getting just a single sample T level – it is secreted in pulses (most in am) so pooling 2-3 samples is best for best correlations (https://pubmed.ncbi.nlm.nih.gov/8676183/) Make sure the samples are done in the am (per your methods)
--	--

VERSION 1 – AUTHOR RESPONSE

Reviewer: 1

Dr. Cvita Gregov, University of Zagreb

Comments to the Author:

The protocol of the study is very interesting and hopefully useful in determining the influence of psychosocial and physical loads on injuries and illnesses in athletes' dual career.

I congratulate the authors and have no further comments on the manuscript's content.

Reviewer: 2

Christian Bjørndal

Comments to the Author:

Thank you for the opportunity to review your study protocol.

I believe your study show potential to become a significant contribution to the research

Reviewer: 3

Dr. Bruce Rogers , University of Central Florida Comments to the Author:

Intro

Line 78, remove “although”, otherwise it does not make sense.

Line 116 – rate to ratings

RESPONSE: Dear Reviewer, thank you for your comments, we amended it as recommended.

Stats

Make sure data is normally distributed to qualify for those selected tests. If not normal an entirely different set of comparisons are needed. With Bland Altman analysis, the differences need to be normally distributed. You should also check for proportional bias using regression-based calculation of mean differences and limits of agreement (Ludbrock, Clin. Exp. Pharmacol. Physiol. 2010, 37, 143–149).

RESPONSE: Thank you for your suggestion, we have amended it accordingly.

The issue of “free testosterone”

Unless you are measuring free testosterone by equilibrium dialysis (doubtful)

I would recommend you measure total T and SHBG per

most guidelines (<https://onlinelibrary.wiley.com/doi/10.1111/cen.13888> and <https://www.nature.com/articles/s41443-019-0144-9>). You will still be able to derive a “calculated” FT

(<http://www.issam.ch/freetesto.htm>).

Additional problems with getting just a single sample T level – it is secreted in pulses (most in am) so pooling 2-3 samples is best for best correlations (<https://pubmed.ncbi.nlm.nih.gov/8676183/>)

Make sure the samples are done in the am (per your methods)

RESPONSE: Thank you for your recommendations and useful literature. We are indeed measuring total T, SHBG and albumin to calculate free testosterone levels. Moreover, two blood samples are taken for each athlete at the single measurement point, in the morning period (7-10 a.m.).

VERSION 2 – REVIEW

REVIEWER	Rogers , Bruce University of Central Florida, Internal Medicine
REVIEW RETURNED	01-Feb-2023
GENERAL COMMENTS	Thank you for your thoughts and good luck